# T Cell Immunoglobulin and Mucin Domain 3 (TIM-3) in Cutaneous Melanoma: A Narrative Review

**DOI:** 10.3390/cancers15061697

**Published:** 2023-03-10

**Authors:** Gerardo Cazzato, Eliano Cascardi, Anna Colagrande, Teresa Lettini, Alessandra Filosa, Francesca Arezzo, Carmelo Lupo, Nadia Casatta, Vera Loizzi, Cristina Pellegrini, Maria Concetta Fargnoli, Eugenio Maiorano, Gerolamo Cicco, Roberto Tamma, Giuseppe Ingravallo

**Affiliations:** 1Section of Molecular Pathology, Department of Precision and Regenerative Medicine and Ionian Area (DiMePRe-J), University of Bari “Aldo Moro”, 70124 Bari, Italy; 2Department of Medical Sciences, University of Turin, 10124 Turin, Italy; 3Pathology Unit, FPO-IRCCS Candiolo Cancer Institute, Str. Provinciale 142 lm 3.95, 10060 Candiolo, Italy; 4Pathology Department, “A. Murri” Hospital-ASUR Marche, Aree Vaste n. 4 and 5, 63900 Fermo, Italy; 5Obstetrics and Gynecology Unit, IRCSS Giovanni Paolo II, 70124 Bari, Italy; 6Innovation Department, Diapath S.p.A., Via Savoldini n.71, 24057 Martinengo, Italy; 7Dermatology, Department of Biotechnological and Applied Clinical Sciences, University of L’Aquila, 67100 L’Aquila, Italy; 8Department of Translational Biomedicine and Neurosciences, Section of Human Anatomy and Histology, School of Medicine and Surgery, University of Bari “Aldo Moro”, Piazza Giulio Cesare, 11 Polyclinic, 70124 Bari, Italy

**Keywords:** TIM-3, T cell immunoglobulin and mucin domain 3, Immunocheckpoint inhibitors, immunotherapy, HMGB1, melanoma

## Abstract

**Simple Summary:**

T cell immunoglobulin and mucin domain 3 (TIM-3) is an inhibitory immunocheckpoint expressed on cytotoxic CD8+ T lymphocytes, NK cells, and myeloid lineage cells in addition to Th1 lymphocytes. Various studies have investigated its role within the tumour microenvironment of melanoma, but also within the melanocytic component, with sometimes conflicting results. In this review, we address the most up-to-date knowledge about this molecule in the context of melanoma, attempting to outline future prospects and potential applications.

**Abstract:**

T cell immunoglobulin and mucin domain 3 (TIM-3) is an inhibitory immunocheckpoint that belongs to the TIM gene family. Monney et al. first discovered it about 20 years ago and linked it to some autoimmune diseases; subsequent studies have revealed that some tumours, including melanoma, have the capacity to produce inhibitory ligands that bind to these receptor checkpoints on tumour-specific immune cells. We conducted a literature search using PubMed, Web of Science (WoS), Scopus, Google Scholar, and Cochrane, searching for the following keywords: “T cell immunoglobulin and mucin-domain containing-3”, “TIM-3” and/or “Immunocheckpoint inhibitors” in combination with “malignant melanoma” or “human malignant melanoma” or “cutaneous melanoma”. The literature search initially turned up 117 documents, 23 of which were duplicates. After verifying eligibility and inclusion criteria, 17 publications were ultimately included. A growing body of scientific evidence considers TIM-3 a valid inhibitory immuno-checkpoint with a very interesting potential in the field of melanoma. However, other recent studies have discovered new roles for TIM-3 that seem almost to contradict previous findings in this regard. All this demonstrates how common and valid the concept of ‘pleiotropism’ is in the TME field, in that the same molecule can behave completely or partially differently depending on the cell type considered or on temporary conditions. Further studies, large case series, and a special focus on the immunophenotype of TIM-3 are absolutely necessary in order to explore this highly promising topic in the near future.

## 1. Introduction

T cell immunoglobulin and mucin domain 3 (TIM-3) is an immunological checkpoint or co-inhibitory receptor. First discovered by Monney et al. in 2002, it is a member of the TIM gene family, located on chromosome 5q33 [1,2]. TIM-3 comprises type I membrane glycoproteins that share a very similar structure consisting of several domains: a variable immunoglobulin (IgV) domain with a stalk consisting of glycosylated mucins of different lengths in the extracellular portion. It has also been expressed by Th 17, cytotoxic CD8+ T lymphocytes, NK cells, and myeloid lineage cells [3,4]. In addition, some authors have demonstrated its existence on developed Th1 lymphocytes, associating it with specific autoimmune-type disorders [2].

On the other hand, it has been realised for some years that various types of neoplasms are capable of producing inhibitory function ligands that bind to these receptor checkpoints on tumour-specific immune cells, inhibiting their functions and allowing the tumour to bypass immunosurveillance [5]. In turn, TIM-3 possesses four potential ligands that can interact with the IgV domain of TIM-3; these are galectin-9 (Gal-9), a nuclear and plasma glycoprotein involved in signal transduction processes and numerous aspects of tumour progression [6]; HMGB1, a molecule released by necrotic/suffering cells, the prototype of Damage Associate Molecular Patterns (DAMPs) [7]; the carcinoembryonic antigen cell adhesion molecule or Ceacam-1, predominantly expressed by activated T lymphocytes [8] and, finally, phosphatidylserine (PtdSer) expressed on the apoptotic cells membranes [8]. These four molecules, which function as TIM-3 ligands, can be found on the surface of T lymphocytes, tumour cells, or necrotic cells [1]. When TIM-3 binds to its ligands, T cells are inhibited, decreasing Th1- and Th17-mediated responses [3,8], as well as CD8+ T lymphocyte proliferation and cytokine production [3]. Gal-9, the first TIM-3 ligand to be identified, specifically binds to a glucidic motif on TIM-3 IgV in antigen-specific CD4 + Th1 cells, inducing and triggering their apoptosis. Additionally, Gal-9 expression on Treg lymphocytes is essential for the inhibition of the TIM-3+ T-lymphocyte effector function [8]. In essence, TIM-3 produced on dendritic cells (DCs) binds to HMGB1, blocking activation [3]. However, it has been proposed that the HMGB1/TIM-3 complex may potentially directly decrease T cell responses by binding to Treg CD8+ (a heterogeneous population consisting of lymphoid cells that express surface markers as FOXP3+, CD122+, CD28low/−, CD45RClow, CD8αα homodimer and Qa-1-restricted, depending on their inhibition activity and the microenvironment they are found in), which in turn inhibits the proliferation of effector T cells [1,9,10]. Therefore, innate immune cells and T cells would undergo HMGB1-induced inhibition through TIM-3-dependent pathways. [1]. TIM-3 would also have a specific inhibitory role in antitumour immunity: it is, indeed, widely expressed on tumour antigen-specific T lymphocytes, both at the peripheral blood level and at the level of tumour-infiltrating lymphocytes (TILs), either CD4+ or CD8+ [11]. Its overexpression characterizes so-called exhausted T cells (especially CD8+, but also CD4+, and particularly, among the latter, CD4+ Treg Foxp3+, but also Th17) in the setting of tumours [12]. Its expression is very high in the exhausted T lymphocytes commonly found in the tumour infiltrate. T cell dysfunction is a process of gradual loss of function that occurs hierarchically during the development of a neoplasm [11,12] when T lymphocytes are persistently exposed to an antigen (tumoral or viral) or differentiation factor (e.g., IL12) [1]. Tim3 + PD1 + CD8+ T cells in the tumour infiltrate show defective production of IFN-γ, TNFα, and IL2 and, in addition, are more likely to remain blocked at the G0 stage of the cell cycle [1,11]. Consequently, high levels of TIM-3 on CD8+ T lymphocytes have been associated with tumour progression and a worse prognosis [1,12]. 

TIM-3 also possesses an inhibitory role on NK cells, inhibiting their cell-mediated cytotoxicity and cytokine production [13], dependent on binding to HMGB1, which actively competes with nucleic acids for binding to TIM-3 [14,15]. Dendritic cells, therefore, being antigen-presenting cells, orchestrate both immunotolerance and the immune response. Through the expression of so-called pattern recognition receptors (PRRs), such as Toll-like receptors (TLRs) and RAGE, which recognize DAMPs released from the distressed cells (of which HMGB1 is the prototype), dendritic cells play an essential role in initiating the immune response against tumours [14]. The HMGB1/TIM-3 complex prevents the HMGB1-mediated recruitment of nucleic acids within endosomes, leading to a decreased efficacy of the immune response and blocking the activation of the immune system [15,16]. The expression of TIM-3 on the surface of DCs can be induced by the release of immunosuppressive factors (such as IL-10) by the neoplasm, which generally occurs in the late stages of tumoral development [16]. 

In addition, TIM-3 promotes the polarisation of macrophages in the M2 subpopulation, which inhibits inflammatory processes and, instead, stimulates tumour processes [17]. Thus, the immune response remains one of the most important factors influencing the development, survival, and expansion of neoplastic cells [16]. In patients with neoplasms, there is certainly a dysfunction of cell-mediated immunity with a consequent reduction of antitumour immunity, this being the arm of the immune system that constitutes the main antitumour mechanism, particularly by CD8+ cytotoxic T lymphocytes [17]. TIM-3, being widely expressed on both T and NK cells, acts directly on this arm, but by performing its functions on DCs, monocytes/macrophages, and endothelial cells, it can exert its inhibitory and pro-tumour function on the whole immune system. Indeed, it directly inhibits cell-mediated immunity by causing the depletion of T lymphocytes and indirectly, both by promoting the proliferation of myeloid-derived suppressor cells (MDCSs) [18] and through its action on cells of innate immunity. High TIM-3 expression, therefore, is correlated with a poor prognosis in cancer patients, as it is one of the most important molecules mediating T cell depletion [19]. The expression of Tim-3 is not restricted to immune cells but is also to tumour cells, and this wide expression corroborates its role in tumour evolution [20].

Cutaneous Melanoma (CM) is the skin cancer with the highest rate of aggressiveness, and incidence, and steadily increasing prevalence rates have been shown in recent years [21]: for example, the estimated CM burden in the USA was 106,110 cases and 7180 deaths in 2021 [21]. Correct recognition of the disease along with accurate staging is vital for the proper management of the diagnostic–therapeutic and care pathway of the CM patient, and diagnostic delays are a major cause of reduced survival rates [22,23,24,25]. In recent times, immunotherapy applied to cases of advanced CM has radically changed the natural history of this neoplasm. In fact, Ipilimumab (anti-CTLA-4 antibody) was approved by the Food and Drug Administration (FDA) back in 2011 for cases of metastatic stage CM. Following data that were not very satisfactory (only 20–30% of advanced disease patients achieved long-term survival with this drug, but with significant toxicity), the clinical use of anti-PD1/PDL-1 therapies was approved and implemented. In any case, in addition to being burdened by toxicity, these therapies do not fully achieve therapeutic goals in all MM patients [5,26]. On this basis, research has been ongoing to find new molecules and receptors that may improve outcomes, and in this field, a role of some importance has been attributed to TIM-3. In this narrative review, we address and discuss the knowledge gained about the role of TIM-3 in melanoma, highlight strengths and weaknesses, and attempt to chart and outline future possible and potential perspectives.

## 2. Materials and Methods

A literature search was conducted on PubMed, Web of Science (WoS), Scopus, Google Scholar, and Cochrane, using following keywords: “T cell immunoglobulin and mucin-domain containing-3”, “TIM-3 and/or “Immunocheckpoint inhibitors” in combination with “malignant melanoma” or “human malignant melanoma” or “cutaneous melanoma”. The time range of our search was until 18 January 2023, and only manuscripts in English were considered without any restriction related to article type. The inclusion criteria stipulated that the included papers should deal with TIM-3 in melanoma, not in general, with a focus on basic, clinical/experimental, and oncological research aspects. Any article that did not deal with the topic of TIM-3 in melanoma was excluded, as were conference abstracts and book paragraphs.

The papers included were evaluated by two authors (G.C. and A.C.) in a double-blind fashion using the techniques advised by the National Institutes of Health Quality Assessment Tool for Case Series Studies. Conflicts were settled by consensus-based discussion; where necessary, a third author (G.I.) was consulted to settle disputes.

## 3. Results

The literature search initially turned up 117 documents, 23 of which were duplicates. Ultimately, 17 publications were included after verifying eligibility and inclusion criteria (Figure 1). Original articles made up the majority of publications (n = 15), followed by reviews (n = 2). According to the 2011 criteria published by the Oxford Centre for Evidence-Based Medicine, every study considered was given a level 4 or 5 confidence rating for clinical research.

## 4. Discussion

In the field of immunotherapy applied to cancer, immunocheckpoint inhibitors play a major role [24,27], and these innovations have also involved the field of application related to CM. In one of the earliest works dating back to 2007, Wiener et al. [28] conducted an elegant study on the presence of mast cells in the tumour microenvironment of MM. Specifically, the authors conducted an analysis of gene expression changes in TGF-βI-treated human mast cells using DNA microarrays. They identified 45 genes that were differentially regulated, including the gene coding for TIM-3. In carrying out this study, one of the previously mentioned TIM-3 ligands, galectin-9, was studied not only at the level of tumour cells but also on mast cells. This study demonstrated that TIM-3 positive mast cells were present in melanoma tissue sections and that neoplastic melanocytes also produced this protein. Additionally, TIM-3 was expressed at a higher level in WM35 and HT168-M1 melanoma cell lines than in isolated epidermal melanocytes, which may have contributed to the tumour cells’ reduced ability to adhere to other cells. A few years later, in 2013, Baghdadi et al. [29] published a study in which they attempted to analyse the TME of melanoma in mouse models, trying to induce long-lasting responses. T cell immunoglobulin mucin-4 (TIM-4) and TIM-3 monoclonal antibodies (mAbs) were specifically used by the authors to increase the therapeutic effects of immunisation against existing B16 murine melanomas. In addition, the combination of anti-TIM-3 and anti-TIM-4 mAbs significantly increased the vaccine-induced antitumour responses against established B16 melanoma. This result was applicable to vaccination using B16 melanoma cells that had been radio-irradiated and modified to express the flt3 ligand gene (FVAX). The authors reported that blocking TIM-3 mainly stimulated antitumour effector activities through natural killer (NK) cell-dependent mechanisms, whereas CD8(+) T cells served as the main effectors induced by the anti-TIM-4 mAb. This evidence was also beginning to emerge from other research groups, such as Fourcade et al. [30]. After demonstrating that in patients with advanced melanoma the NY-ESO-1 antigen stimulated spontaneous NY-ESO-1-specific CD8(+) T cells, they demonstrated through the analysis of CD8(+) T-lymphocytes that a subset of these PD-1(+) NY-ESO-1-specific CD8(+) T cells regulated Tim-3 expression and that Tim-3(+)PD-1(+) NY-ESO-1-specific CD8(+) T cells were more dysfunctional than Tim-3(-)PD-1(+) and Tim-3(-)PD-1(-) NY-ESO-1. As a result, NY-ESO-1-specific CD8(+) T cells produced more cytokines after brief ex vivo stimulation with the corresponding peptide when Tim-3-Tim-3L was blocked, increasing the functional potential of these cells. Therefore, Tim-3-Tim-3L blockade increased cytokine generation and the proliferation of CD8(+) T cells specific for NY-ESO-1 after extended antigen stimulation, which worked in concert with blocking PD-1-PD-L1. The idea of blocking Tim-3-Tim-3L in conjunction with PD-1-PD-L1 blockade to correct tumour-induced T cell depletion/dysfunction in patients with advanced melanoma was further supported by this study. In 2014, Da Silva et al. [31] concentrated on the expression of TIM-3 on natural killer cells, which have the ability to constitutively express Tim-3. The authors compared the function of Tim-3 in NK cells from healthy donors and patients with metastatic melanoma in this study. The patients’ NK cells were functionally compromised/tired, and the exhausted phenotype was reversed when TIM-3 blockers were administered. The authors also showed, for the first time, a correlation between Tim-3 expression levels and disease stage and poor prognostic variables.

It is also important to mention Wu Feng-Wua et al. [32], who focused in their study on the mediating potential of TIM-3 in carcinogenesis. In particular, the authors found that Tim-3, in addition to being expressed on the cell types previously considered, can also be expressed on endothelial cells after stimulation with Toll-Like Receptor 4 (TLR4) ligand released by tumour cells. In this context, Tim-3 expressed by endothelial cells did not function as a galectin-9 receptor (as previously shown) but was actually shown to mediate the interaction of endothelial cells with tumour cells. The work showed that this docking at the level of B16 melanoma cells could trigger the NF-kappaB signalling pathway, the activation of which not only promoted cell proliferation but also increased resistance to apoptosis through the up-regulation of factors such as Bcl-2 and Bcl-xL and the down-regulation of Bax. Thus, Tim-3 was shown to facilitate the survival of B16 cells in the bloodstream, which were arrested in the lung and, upon invading, caused more metastatic nodules in the lung. These results suggested that Tim-3 expressed by endothelial cells increased the metastatic potential of tumour cells by facilitating intravasation, survival in the blood stream, and extravasation of tumour cells. Patel et al. [33] were credited with studying and analysing the expression of TIM-3 on a particular element present in the TME, namely tumour-associated dendritic cells (TADCs). Specifically, the authors showed that TIM-3 was predominantly expressed by TADCs and interacted with HMGB1, suppressing nucleic acid-mediated activation (preventing localisation in endosomal vesicles) of an effective anti-tumour immune response. Although the work cited and analysed so far considers TIM-3 as a marker of immunosuppression in TME, a very recent paper by Schatton et al. [34] identified an additional role for Tim-3 as a growth suppressive receptor intrinsic to melanoma cells. According to the authors, overexpression of Tim-3, specific to melanoma, inhibited carcinogenesis, whereas Tim-3 inhibition of melanoma cells enhanced tumour growth in both immunocompetent and immunocompromised animals. The growth of immunogenic murine melanomas in T-competent hosts was particularly decreased by blocking Ab-mediated Tim-3, consistent with the known antitumour effects of T cell-mediated Tim-3 inhibition. The growth-promoting effects of the melanoma-Tim-3 antagonistic relationship, however, were confirmed when Ab injection of Tim-3 increased the carcinogenesis of both highly and poorly immunogenic murine and human melanomas in T cell-deficient animals. The data demonstrated that downstream pro-proliferative MAPK signalling mediators phosphorylation was decreased by melanoma-Tim-3 activation and elevated by its inhibition. Finally, in T cell-deficient mice, pharmacological inhibition of MAPK reversed unintended Tim-3 Ab-mediated tumorigenesis, while in T cell-competent hosts, Tim-3 interference boosted the desirable anti-tumour action. These results identified the Tim-3 blockade of melanoma as a mechanism that antagonises the therapeutic efficacy of Tim-3-directed T cells. Additionally, they demonstrated that MAPK targeting could be a combined strategy to avoid the negative effects of accidental melanoma-Tim-3 inhibition. Liu et al. [35] presented data on the combination of the extracellular signal-regulated mitogen-activated kinase (MEK) inhibitor, trametinib, which has already demonstrated efficacy in patients with advanced melanoma, and Tim-3. They worked under the presumption that the role of the mitogen-activated protein kinase (MAPK) pathway and immune checkpoints could make targeted therapy combined with immunotherapy a promising regimen. The combination of trametinib and anti-Tim-3 monoclonal antibody (mAb) in the treatment of B16-F10 melanoma mice led to the observation that trametinib significantly promoted apoptosis and inhibited cell proliferation. At the same time, MEK inhibition enhanced Tim-3 expression and caused a decrease in CD8+ T cells. Conversely, the anti-Tim-3 antibody increased anti-tumour immunity by stimulating CD8+ T cells, so the combined therapy produced a potent anti-tumour effect in a cooperative manner. Another recent study by Li et al. [36] aimed to identify the physiological mechanisms by which TIM-3 contained in exosomes from melanoma cells might regulate CD4+ T cell immunological activity and macrophage polarisation in the M2 phenotype in MM. The authors examined the expression pattern of TIM-3 using exosomes obtained from the human melanoma cell line MV3 in an effort to achieve this. Secondly, to evaluate the polarisation of macrophages toward the M2 phenotype and to measure the amounts of several variables relevant to the Epithelial Mesenchymal Transition, exosomes from MV3 cells modified with sh-TIM-3 were co-incubated with CD4+ T cells (EMT). It is important to note that silencing TIM-3 enhanced the immune function of CD4+ T cells and inhibited M2 macrophage polarisation, attenuating the growth and metastasis of melanoma cells. Overall, TIM-3 loaded with exosomes derived from MV3 cells suppressed CD4+ T cell immune functions and induced M2 macrophage polarisation to promote tumorigenesis and, subsequently, disease progression. This work has the merit of laying a foundation for the development of future molecular therapies.

Finally, Pagliano et al. [37] evaluated the effects of Tim-3 blockade in a murine model and human patients, showing that PD-1+CD8+ Tim-3+ TILs upregulate phosphatidylserine, a ligand for Tim-3, and are capable of acquiring and expressing cell surface myeloid markers from antigen-presenting cells through the transfer of membrane fragments, a process known as “trogocytosis” In their studies, inhibiting Tim-3 expression on APCs that expressed it prevented CD8+ T cells and PD-1+Tim-3+ CD8+ TILs isolated from melanoma patients from going into trogocytosis. Tim-3 and PD-1 synergistically blocked CD8+ TIL trogocytosis in two mouse melanoma models, lowering the tumour burden and extending life. Tim-3 deletion reduced CD8+ TIL trogocytosis in dendritic cells but not in CD8+ T cells.

Finally, although not directly identified in the literature review, it is important to emphasise certain concepts in relation to TIM-3 immunoexpression by immunohistochemistry. In particular, as yet, there are no studies that have evaluated TIM-3 expression in the context of CM, although there are studies that have investigated it in other backgrounds. For example, in a 2019 paper, Wang J.J. et al. performed a retrospective study of nine melanoma patients who had developed intracranial metastases. The authors subjected the extracranial and intracranial tumour specimens to immunophenotypic staining for programmed death ligand 1 (PD-L1), programmed cell death receptor 1 (PD-1), lymphocyte activation gene-3 (LAG-3) and T cell immunoglobulin and domain-mucin containing-3 (TIM-3). TIM-3 presented 33.33% immunohistochemical staining in the TME of both primary tumour sites and metastases, whereas it did not present staining of neoplastic cells. Correlating the expression of TIM-3 with that of the other three molecules, the authors postulated that the expression of TIM-3 could corroborate to maintain an immunosuppressive stage responsible for the disease progression observed in these patients [38]. In another recent paper by Conway J.W. et al., an attempt was made to study the TME of usually metastatic sites in stage IV CM. In more detail, the authors analysed the density, spatial distribution, and population subtypes of immune cells from the liver, lung, brain, subcutaneous, and lymph node tissues of 130 stage IV patients. Correlating the sites of metastasis with OS, the authors showed that a location of CM in the liver, bone, or brain was more predictive of reduced OS than the location in the lung and lymph nodes. Furthermore, it appeared that T lymphocytes from CM liver localisations expressed lower levels of PD-1 compared to other sites and increased levels of TIM-3 expression compared to lymph node, brain, and subcutaneous metastases. Hence, the authors suggested that immunotherapy targeting TIM-3 in the subgroup of patients with stage IV liver metastases might be a promising tool for the treatment of these subjects [39].

On the other hand, it is interesting to note that there are no studies that have tried to analyse TIM-3 immunoexpression directly on melanoma cells, and some of our preliminary results (Figure 2), not yet published, would seem to indicate a differential expression at the level of CM neoplastic cells.

## 5. Conclusions

Immunotherapy for cancer has revolutionised the landscape of medical oncology and immuno-oncology in just a few years. A growing body of scientific evidence considers TIM-3 a valid inhibitory immunocheckpoint with a very interesting potential in the field of melanoma. On the other hand, other recent studies have discovered new roles of TIM-3 that seem almost to contradict previous findings in this regard. All this demonstrates how common and valid the concept of ‘pleiotropism’ is in the TME field, in that the same molecule can behave completely or partially differently depending on the cell type considered or on temporary conditions. Therefore, further studies, large case series, and a special focus on the immunophenotype of TIM-3 are absolutely necessary in order to explore this highly promising topic in greater detail in the near future.

## Figures and Tables

**Figure 1 cancers-15-01697-f001:**
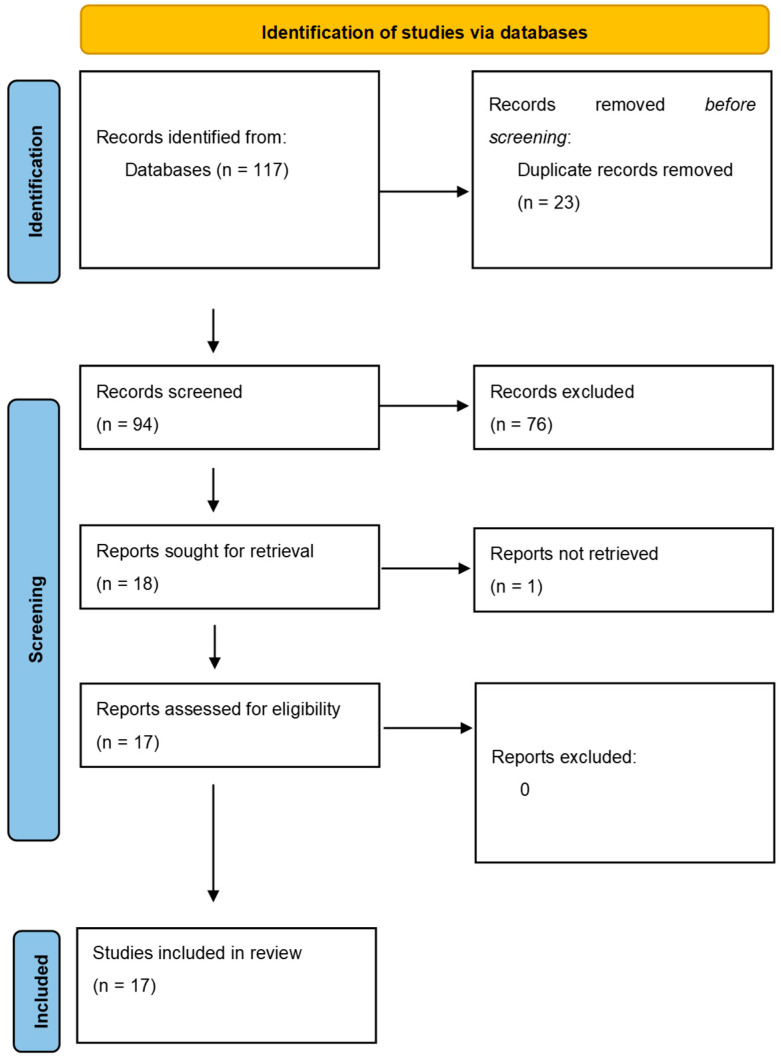
Articles selection and flowchart of the review.

**Figure 2 cancers-15-01697-f002:**
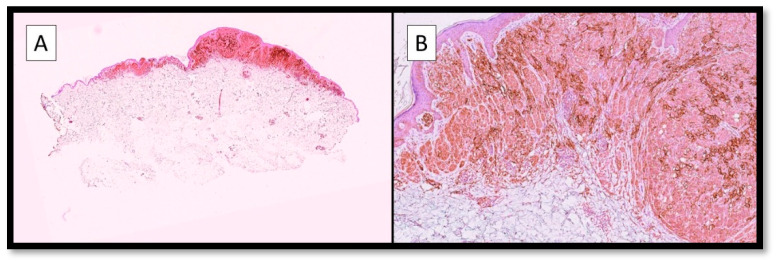
(**A**) Photomicrograph showing an example of immunostaining for TIM-3 in a malignant melanoma specimen. Note the broad positivity of the brown signal at the level of both the neoplastic cells and the TME (Immunostaining for TIM-3, Original Magnification 4×). (**B**) Immunohistochemical preparation for TIM-3: note, in more detail, the strong and widespread cytoplasmic positivity of melanoma cells for TIM-3, as well as the presence of positive cells in the TME. Although no data is available in the literature yet, and our data have not yet been published, this expression pattern fits well with the concept of ‘pleiotropism’ specific to TIM-3 (Immunostaining for TIM-3, Original Magnification 10×).

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
