# Peer review of "T Cell Immunoglobulin and Mucin Domain 3 (TIM-3) in Cutaneous Melanoma: A Narrative Review"

_cancers, 2023, doi:10.3390/cancers15061697_

Round 1
Reviewer 1 Report
The authors review the current understanding of T cell immunoglobulin and mucin domain 3 (Tim-3) function in melanoma pathogenesis and treatment. Their literature search yielded 117 manuscripts, and ultimately, 17 publications were reviewed. The review is timely and represents a good overview of the current state of the field. Below are a few minor comments:
1. I would appreciate a better description of the inclusion/exclusion criteria and the potential conflicts that arose necessitating a third author to arbitrate. A seemingly large number (76) publications were excluded, and the reasons for exclusion was vague.
2. Line 62 – I may consider broadening the language a bit to include more CD4+ T cell subsets as Tim-3 is expressed on T regulatory, Th1, and Th17 cells.
3. Line 84 – Given the controversy surrounding the existence of T regulatory CD8+ cells, I may soften this language.
4. I think that much of the discrepant reports of Tim-3 function in melanoma stem from the various functions of Tim-3 on different cell types present in the tumor microenvironment. This could be stated more explicitly in text and/or figure form.
Author Response
Reviewer n’1: The authors review the current understanding of T cell immunoglobulin and mucin domain 3 (Tim-3) function in melanoma pathogenesis and treatment. Their literature search yielded 117 manuscripts, and ultimately, 17 publications were reviewed. The review is timely and represents a good overview of the current state of the field. Below are a few minor comments:
Answer n’1: Dear Reviewer n’1 thank you very much for this wonderful words.
Reviewer n’1: 1. I would appreciate a better description of the inclusion/exclusion criteria and the potential conflicts that arose necessitating a third author to arbitrate. A seemingly large number (76) publications were excluded, and the reasons for exclusion was vague.
Answer n’2: We have added a sentence testifying what the inclusion and exclusion criteria were and, in addition, we confirm that only for a few papers (5) it was necessary to consult a third reviewer, as it was not clear whether they also referred to melanoma. Thank you again.
Reviewer n’1: 2. Line 62 – I may consider broadening the language a bit to include more CD4+ T cell subsets as Tim-3 is expressed on T regulatory, Th1, and Th17 cells.
Answer n’3: Done dear Reviewer n’1. Thank you very much.
Reviewer n’1: 3. Line 84 – Given the controversy surrounding the existence of T regulatory CD8+ cells, I may soften this language.
Answer n’4: Dear Reviewer n’1, thank you very much. We corrected this language.
Reviewer n’1: 4. I think that much of the discrepant reports of Tim-3 function in melanoma stem from the various functions of Tim-3 on different cell types present in the tumor microenvironment. This could be stated more explicitly in text and/or figure form.
Answer n’5: Dear Reviewer n’1, done. Thank you very much. We added a sentence in Conclusion section about the pleiotropism effect of TIM-3 in TME. We hope that paper will be fine now. Thanks again for your help.
Reviewer 2 Report
This review article summarizes the current literature on the role of Tim3 in human cutaneous melanoma. The title is extremely specific however the text is not, given that half of the papers cited involves murine studies we would suggest that the authors revise the title in this regard. In conclusion, the discussion is well balanced however the paper is a bit difficult to read in the present form we have some suggestions listed below that will improve the quality of the manuscript. Additionally, appropriate references should be included.
Major comments:
Overall, the idea of T cell exhaustion or dysfunction is not clearly indicated or even misleading at some points. Line 92-93 “T-lymphocyte exhaustion is directly involved in the loss of an immunosuppressive state against cancer” is misleading.
Although T cell dysfunction may ultimately lead to depletion, we would recommend the authors to not use these terms interchangeably.
“TIM-3+ PD-1+depleted T lymphocytes” line 98-99 is incorrect. Please rephrase as Tim3+PD1+CD8+ T cells.
Please cite a paper describing human cancer for line 102. Reference 12 is murine study.
Minor comments:
1) What does “2 and “4” indicate in the abstract. Are those bullet-points or references. Where is 1 and 3?
2) The first reference is 2, where is reference 1?
3) In line 56, please correct the “Immunoglobulin and T-cell mucin domain 3 (Tim3) to T cell immunoglobulin and mucin domain 3”.
4) Tim3 was discovered in 2002, please correct information in line 58.
5) Please breakdown the sentences in lines 58 to 63
6) Line 63-64 indicates several publications however there is citation for only 1.
7) Line 84, please define Treg CD8+. Please indicate the primary paper showing these findings.
8) Please cite the original paper for line 98
9) Please correct line 113
10)Please rephrase line 213-214
11)Line 227, please refrain from using “clinical response” in murine model.
12)Line 321 please rephrase as “in murine model and human patients” instead of “in murine and human models”
Author Response
Reviewer n’2: This review article summarizes the current literature on the role of Tim3 in human cutaneous melanoma. The title is extremely specific however the text is not, given that half of the papers cited involves murine studies we would suggest that the authors revise the title in this regard. In conclusion, the discussion is well balanced however the paper is a bit difficult to read in the present form we have some suggestions listed below that will improve the quality of the manuscript. Additionally, appropriate references should be included:
Answer n’1: Dear Reviewer n’1 thank you very much for these words useful to improve the quality of our manuscript. We will follow all your suggestions. So, we changed the title including melanoma, not only human.
Reviewer n’2: Overall, the idea of T cell exhaustion or dysfunction is not clearly indicated or even misleading at some points. Line 92-93 “T-lymphocyte exhaustion is directly involved in the loss of an immunosuppressive state against cancer” is misleading. Although T cell dysfunction may ultimately lead to depletion, we would recommend the authors to not use these terms interchangeably .
Answer n’2: Ok, thank you very much for this tip. It’s right. So, we changed these sentences.
Reviewer n’2: “TIM-3+ PD-1+depleted T lymphocytes” line 98-99 is incorrect. Please rephrase as Tim3+PD1+CD8+ T cells. Please cite a paper describing human cancer for line 102. Reference 12 is murine study.
Answer n’3: Done dear Reviewer n’2. Thank you very much.
Reviewer n’2: 1) What does “2 and “4” indicate in the abstract. Are those bullet-points or references. Where is 1 and 3?.
Answer n’4: Dear Reviewer no. 2, sorry for this typo. actually, in the process of deleting the numbers for the paragraphs in the abstract, we forgot to delete the number 2 and 4.
Reviewer n’2: 2) The first reference is 2, where is reference 1?
Answer n’5: Dear Reviewer No. 2, sorry for this mistake. We have corrected it.
Reviewer n’2: In line 56, please correct the “Immunoglobulin and T-cell mucin domain 3 (Tim3) to T cell immunoglobulin and mucin domain 3”.
Answer n’6: Dear Reviewer n’2, thank you very much. We corrected it.
Reviewer n’2: Tim3 was discovered in 2002, please correct information in line 58.
Answer n’7: Done.
Reviewer n’2: Please breakdown the sentences in lines 58 to 63.
Answer n’8: Done, thank you very much.
Reviewer n’2: Line 63-64 indicates several publications however there is citation for only 1.
Answer n’9: ok. We have corrected. Thank you.
Reviewer n’2: Line 84, please define Treg CD8+. Please indicate the primary paper showing these findings.
Answer n’10: Done. Thank you.
Reviewer n’2: Please cite the original paper for line 98
Answer n’11: Done. Thank you.
Reviewer n’2: Please correct line 113.
Answer n’12: Done, thank you.
Reviewer n’13: Please rephrase line 213-214.
Answer n’13: Done.
Reviewer n’14: Line 227, please refrain from using “clinical response” in murine model.
Answer n’14: ok, we have corrected.
Reviewer n’15: Line 321 please rephrase as “in murine model and human patients” instead of “in murine and human models”
Answer n’15: Done. Thank you very much.
Round 2
Reviewer 2 Report
The authors have made most of changes suggested. However, some further suggestions are as follows:
1) Title: T cell immunoglobulin and mucin domain 3 (TIM-3) in Cutaneous Melanoma: a narrative review
2) Line 82 please define the “certain markers”.
3) As pointed out before, please do not use T cell dysfunction or depletion interchangeably (Line 92).
Author Response
Dear Reviewer n'2,
thank you very much
All done.